# Regulation of Antigenic Variation by *Trypanosoma brucei* Telomere Proteins Depends on Their Unique DNA Binding Activities

**DOI:** 10.3390/pathogens10080967

**Published:** 2021-07-30

**Authors:** Bibo Li, Yanxiang Zhao

**Affiliations:** 1Center for Gene Regulation in Health and Disease, Department of Biological, Geological, and Environmental Sciences, College of Sciences and Health Professions, Cleveland State University, 2121 Euclid Avenue, Cleveland, OH 44115, USA; 2Case Comprehensive Cancer Center, Case Western Reserve University, 10900 Euclid Avenue, Cleveland, OH 44106, USA; 3Department of Inflammation and Immunity, Lerner Research Institute, Cleveland Clinic, 9500 Euclid Avenue, Cleveland, OH 44195, USA; 4Center for RNA Science and Therapeutics, Case Western Reserve University, 10900 Euclid Avenue, Cleveland, OH 44106, USA; 5Shenzhen Research Institute, The Hong Kong Polytechnic University, Shenzhen, China; 6State Key Laboratory of Chemical Biology and Drug Discovery, Department of Applied Biology and Chemical Technology, The Hong Kong Polytechnic University, Hung Hom, Hong Kong, China

**Keywords:** antigenic variation, *Trypanosoma brucei*, VSG, telomere, TRF, RAP1, Myb domain, DNA binding activity

## Abstract

*Trypanosoma brucei* causes human African trypanosomiasis and regularly switches its major surface antigen, Variant Surface Glycoprotein (VSG), to evade the host immune response. Such antigenic variation is a key pathogenesis mechanism that enables *T. brucei* to establish long-term infections. VSG is expressed exclusively from subtelomere loci in a strictly monoallelic manner, and DNA recombination is an important VSG switching pathway. The integrity of telomere and subtelomere structure, maintained by multiple telomere proteins, is essential for *T. brucei* viability and for regulating the monoallelic VSG expression and VSG switching. Here we will focus on *T. brucei* TRF and RAP1, two telomere proteins with unique nucleic acid binding activities, and summarize their functions in telomere integrity and stability, VSG switching, and monoallelic VSG expression. Targeting the unique features of *Tb*TRF and *Tb*RAP1′s nucleic acid binding activities to perturb the integrity of telomere structure and disrupt VSG monoallelic expression may serve as potential therapeutic strategy against *T. brucei*.

## 1. Antigenic Variation in *T. brucei* Involves Its Major Surface Antigen, Variant SURFACE Glycoprotein (VSG)

*Trypanosoma brucei* is a protozoan parasite that causes human African trypanosomiasis. It is transmitted by tsetse flies (*Glossina* spp.) that inhabit 36 sub-Saharan African countries, thus endangering ~60 million people in this region [1]. Two other very closely related trypanosomes, *Trypanosoma cruzi* and *Leishmania*, cause debilitating Chagas disease and Leishmaniasis in humans, respectively. Together, ~20 million people worldwide are infected by these kinetoplastid parasites [2]. In addition, *T. brucei* can infect livestock, which severely affects the economics in sub-Sahara Africa [3]. However, few drugs are available to treat these diseases safely and effectively with easy administering, and drug resistance cases have been observed [2].

*T. brucei* is heteroxenous and has several key life cycle stages. After ingested by its insect vector during a blood meal on an infected mammal, *T. brucei* proliferates in the tsetse midgut [4]. At this stage, procyclic form (PF) *T. brucei* expresses procyclins as its major surface proteins, as the C-termini of procyclins are resistant to tsetse midgut enzymes [5]. Subsequently, *T. brucei* migrates to tsetse’s proventriculus and then salivary glands, where it differentiates into the metacyclic form (MF) in the latter organ. MF *T. brucei* is not proliferative and expresses VSG as its major surface antigen. When tsetse takes another blood meal, the parasites can be injected into a new mammalian host [6]. After *T. brucei* enters its mammalian host, it differentiates into the bloodstream form (BF), which proliferates in extra cellular spaces of its host and expresses VSG as its major surface antigen [6]. VSG is highly immunogenic [7]. However, *T. brucei* undergoes antigenic variation and sequentially expresses immunologically distinct VSGs, thereby effectively evading the host immune response [8]. Antigenic variation is a key pathogenesis mechanism and is essential for *T. brucei* to establish long-term infections.

### 1.1. VSG Is Expressed in a Monoallelic Manner from a Single VSG Expression Site (ES)

*T. brucei* has a large *VSG* gene pool with >2500 *VSG* genes and pseudogenes [9]. Most of these are located in subtelomeric gene arrays (Figure 1A) [9,10] on the 11 pairs of megabase chromosomes, which are 0.9–5.7 Mb and contain all essential genes [11,12]. In addition, *T. brucei* has ~100 minichromosomes, which are 30–150 kb and mainly include central 177 bp repeats and terminal telomere repeats [12,13]. Individual *VSG* genes can be found on two thirds of the minichromosome telomeres (Figure 1B) [9], contributing to the *VSG* gene pool size. In distinct contrast to the fact that protein-coding genes are transcribed by RNA polymerase II (RNAP II), VSGs are transcribed by RNA polymerase I (RNAP I) [14] exclusively from BF VSG expression sites (ESs) (Figure 1C) while growing in its mammalian host and from MF ESs (Figure 1D) while residing in the tsetse salivary glands. BF VSG ESs are polycistronic transcription units (PTUs) [15] located at subtelomeres (Figure 1C) [16] of megabase chromosomes and, in one case, at the subtelomere of an intermediate chromosome (*T. brucei* has four to five intermediate chromosomes that are 200–700 kb [12]). *VSG* is the last gene in any ES and within 2 kb from the telomere repeats, while the ES promoter is located 40–60 kb upstream [15]. In contrast, MF ES are telomeric monocistronic transcription units with the ES promoter located ~5 kb upstream of the telomere (Figure 1D) [17].

*T. brucei* has multiple ESs (~15 BF ESs in the Lister 427 strain) [10,15], all with the same gene organization and ~90% sequence identity [15]. However, only one ES is fully active, expressing only one type of VSG on the cell surface at any time (Figure 1E) [18]. Monoallelic VSG expression has been shown to depend on multiple factors, although the detailed mechanisms of how it is achieved are still unclear. It has been shown that defects in a telomere protein *Tb*RAP1 [19,20,21,22], chromatin structure [20,23,24,25,26,27,28,29,30,31,32], transcription elongation [33,34,35], the inositol phosphate pathway [36,37], nuclear lamina [38,39], recruitment of sumoylated protein(s) to the active ES promoter [40], DNA replication initiation factors [41,42,43,44], and a subtelomere and VSG-associated VEX complex [45,46,47] can all abolish VSG monoallelic expression. Although monoallelic VSG expression is not essential for ex vivo cultured cells, parasites expressing multiple VSGs are more quickly eliminated by its mammalian host [48], indicating that monoallelic *VSG* expression is important for the parasite virulence.

### 1.2. Several Pathways Mediate VSG Switching

Proliferative BF *T. brucei* undergoes VSG switching regularly [49]. Most VSG switching events are either transcriptional or DNA recombination-mediated (Figure 2) [50,51]. In a transcriptional in situ switch, the originally active ES is silenced while an originally silent ES becomes fully expressed, and no gene rearrangement is involved. Recombination-mediated switches have two major types. In gene conversion events, a silent *VSG* is duplicated into the active ES to replace the active *VSG*, which is subsequently lost. In crossover (or telomere exchange) events, a silent *VSG* (frequently together with its downstream telomere sequence) exchange places with the active *VSG* without any loss of genetic information. In addition, fragments of several silent *VSG* genes can also be recombined to form a new mosaic *VSG* gene in the originally active ES [52,53]. Furthermore, it has been observed that the active ES can simply be lost and a different ES derepressed during a switching event, and such “ES loss + in situ” switches can be frequently detected when telomere proteins are depleted [54,55]. In many recent studies, DNA recombination-mediated switches are much more prevalent than in situ switches [54,55,56,57,58,59,60,61,62,63].

Proteins involved in DNA recombination include RAD51 that mediates strand invasion in homologous recombination (HR) [64], RAD51-3 (a RAD51-related protein) [65], and BRCA2 [66] facilitate VSG switching. Proteins involved in DNA metabolism including Topoisomerase 3 alpha [62]; the RMI1 homolog [59]; a replication origin binding factor *Tb*ORC1 [41]; and a RecQ helicase, RECQ2 [67], suppress VSG switching. Additionally, telomere proteins have been shown to suppress VSG switching [54,55,63,68], while cells harboring a critically short active *VSG*-adjacent telomere have an increased VSG switching rate [61]. Furthermore, cells with defective RNase H enzymes that degrade the RNA strand in the DNA:RNA hybrid appear to have a higher VSG switching frequency than WT cells [69,70].

How VSG switching is initiated and regulated is not clear [71,72]. For recombination-mediated switching events, sequences flanking the *VSG* gene provide the sequence homology. All *VSG* 3′UTRs have the conserved 16 bp and 9 bp motifs with consensus sequences, and 70 bp repeats are located upstream of 70–80% of *VSG* genes [9,73]. The 70 bp repeats and *VSG* 3′UTRs can mediate efficient HR. In addition, telomere repeats are located immediately downstream of all ES-linked *VSGs* and *VSGs* on minichromosomes [9], which can also provide good sequence homology. Since HR is an efficient DNA damage repair mechanism with high fidelity [74], DNA double-strand breaks (DSBs) within the homologous sequences flanking the active *VSG* have been proposed to be a potent VSG switching inducer [71,72]. Indeed, introducing an I-SceI cut between the 70 bp repeats and the active *VSG* gene caused a 250-fold increase in the VSG switching rate [56,57]. In addition, DSBs can be detected at subtelomeres in WT *T. brucei* cells [56]. However, how DSB is naturally induced in the parasite, and whether this is the only initiator for VSG switching still awaits further investigations.

As VSG is expressed exclusively from subtelomeric loci; the telomere and subtelomere integrity has been shown to be a key factor influencing VSG switching frequency. Introducing a DSB immediately upstream of the active *VSG* increased the switching frequency by ~250 fold in vitro with most switching events mediated by DNA recombination [56,57]. In addition, *T. brucei* cells carrying a critically short telomere downstream of the active *VSG* have a VSG switching rate ~10-fold higher than that in cells with normal sized telomeres [61]. Furthermore, telomere proteins have been shown to play important roles in VSG monoallelic expression [19,20,21,22] and affect VSG switching frequencies [21,54,55,63,68]. Depletion of telomere proteins *Tb*TRF, *Tb*RAP1, or *Tb*TIF2 induces more telomere/subtelomere DNA damage, disrupts the telomere/subtelomere stability, and results in significantly increased VSG switching rates [21,22,54,55,63,68,75].

## 2. Telomere Functions in Antigenic Variation

Linear chromosomes in eukaryotic cells are capped by a special nucleoprotein complex called telomere. In most eukaryotes, telomere DNA consists of a simple repetitive TG-rich sequences [76]. Although kinetoplastids branched away from vertebrates more than 500 million years ago during evolution, telomeres in both vertebrates and kinetoplastids including *T. brucei* contain TTAGGG repeats [76]. In most eukaryotes, telomeres end in a 3′ single-stranded overhang structure [77,78,79,80,81,82,83,84,85], which can invade the duplex telomere region and form a T-loop structure that has been observed in human, mouse, chicken, *T. brucei*, ciliates, common garden pea, *C. elegans*, and *K. lactis* [86,87,88,89,90,91,92]. Conventional DNA polymerases cannot fully replicate the ends of linear DNA molecules, resulting in the so-called “end replication problem” [93]. Many eukaryotes use a specialized reverse transcriptase, telomerase, to synthesize the G-rich telomere strand de novo according to a short template provided by the telomerase RNA component [94,95,96,97]. In the absence of the telomerase activity, telomere can be maintained by HR (such as breakage-induced replication) in ~15% cancer cells [98,99] and in telomerase null yeast survivors [100,101].

### 2.1. Telomeres Are Essential for Genome Integrity and Affect Nearby Gene Expression

A key function of the telomere is to mask the natural chromosome ends from being recognized by the DNA damage response machineries as a DNA damage site. Exposed telomeres are not only vulnerable to nucleolytic degradation but can also be processed to form chromosome end-to-end fusions that lead to the “breakage-fusion-bridge” cycle [102], which can induce loss of heterozygosity, non-reciprocal translocations, and gene amplification [103]. Indeed, recent studies in mammalian cells have shown that telomere end fusions can lead to chromothripsis and kataegis [104], while chromoanagenesis (including chromothripsis, chromoplexy, and chromoanasynthesis) is an important mechanism of genome instability that can contribute to tumorigenesis [105,106].

In humans, Shelterin [107]—a complex including six telomere proteins (TRF1 [108], TRF2 [109,110], RAP1 [111], TIN2 [112], TPP1 [113,114,115], and POT1 [116,117]) and the CST complex (CTC1/STN1/TEN1 in vertebrates and CDC13/STN1/TEN1 in budding yeast) [118,119] are key components of the telomere complex that are indispensable for chromosome end protection. TRF1 and TRF2 bind the duplex TTAGGG repeats [108,120,121,122,123] while POT1 [116,124] and the CST complex bind the single-stranded telomere G-overhang [125]. RAP1 interacts with TRF2 [111], and TIN2 interacts with both TRF1 and TRF2 [112,126], while TPP1 interacts with both TIN2 and POT1 [113,114,115]. The Shelterin components can help protect telomere stability by inhibition of Non-Homologous End Joining [127,128,129], HR [130,131,132,133], and Microhomology-Mediated End Joining [134,135] at the telomere. In addition, the T-loop structure buries the telomere G-overhang, which suppresses ATM activation at mammalian telomeres [136].

The telomere complex can also suppress the nearby gene expression [137]. This telomere position effect or telomeric silencing is an epigenetic phenomenon, depending on the telomere heterochromatic structure [138]. Telomeric silencing is best understood in budding yeast, where *Sc*RAP1 is the predominant duplex telomere DNA binding factor [139,140,141]. *Sc*RAP1 recruits SIR3 and SIR4 proteins [142,143,144,145,146], which in turn recruit SIR2, a histone deacetylase [147], to nucleate and maintain the telomere heterochromatic structure [148]. In general, genes located closer to the telomere are more strongly repressed than genes located further away [147]. In addition, longer telomere repeats have stronger telomeric silencing effects in budding yeast [149,150], presumably because more *Sc*RAP1 proteins are recruited to the telomere DNA.

Although telomeres frequently form a heterochromatic structure, the Telomere Repeat-containing RNA (TERRA) has been detected in many organisms including *T. brucei* [63,75,151,152,153], several kinetoplastids and *Plasmodium falciparum* [151,152,154], human [155], mouse [156], fission [157] and budding yeasts [158], and birds [159]. TERRA is prone to form an R-loop structure with the telomere DNA (a three-stranded structure containing a DNA:RNA hybrid and a displaced ssDNA) [160]. Both TERRA and telomeric R-loop (TRL) have been shown to regulate telomerase-dependent and recombination-mediated telomere maintenance and also play a role in chromosome end protection [153,160,161]. *T. brucei* TERRA has a few unique features compared to that in human and yeast cells, where frequently multiple telomeres are transcribed by RNAP II [155,157,158,162,163,164,165,166,167,168]. First, in *T. brucei,* TERRA is transcribed by RNAP I as a read-through product when RNAP I continues into the telomeric repeats downstream of the active *VSG* [63,75,151,152]. Second, the active ES-adjacent telomere is the only TERRA transcription site [75]. Third, the number of nuclear TERRA foci is cell cycle-regulated in *T. brucei* [75]. Most G1 cells (~60%) have only a single TERRA focus, and the number of TERRA foci increases as cells enters S and G2/M stages [75].

### 2.2. DNA Binding Activities of Telomere Proteins Are Critical for Their Essential Functions

Telomere binding proteins apparently play pivotal roles in key telomere functions, as they nucleate the telomere nucleoprotein complex. Studies in yeasts, mammals, plants, ciliates, and kinetoplastids have shown that two major DNA binding activities—the Myb motif-mediated duplex telomere DNA binding [141,169,170] and the OB fold-mediated single-stranded telomere DNA binding [171]—are conserved across many species [172,173]. TRF1 and TRF2 are the first telomere proteins that have been found to bind the duplex TTAGGG repeats with their C-terminal Myb domains [108,109,110]. Frequently, two Myb domains are necessary for a robust DNA binding [174]. This was confirmed to be true in human TRF1/2 proteins [175,176]. Mammalian TRF1 and TRF2 have a TRF Homology (TRFH) domain towards their N-termini that can homodimerize, which allows TRF1 and TRF2 dimers to bind the duplex telomere DNA [108,109,110]. Hence, an elegant set of experiments were performed before the era of RNA interference, TELENs, or CRISPR/cas using dominant negative Myb domain deletion mutants of human TRF1 and TRF2 [175,176]. When overexpressed, Myb deletion TRF mutants tether the endogenous WT TRFs off the telomere through the TRFH-mediated dimerization, as the mutant-WT dimer only possesses a single Myb domain and cannot bind the telomere DNA sufficiently [175,176]. Fission yeast *Sp*TAZ1 that binds the duplex telomere DNA was shown to be a functional homolog of TRF and to also have a C-terminal Myb motif [177,178]. Subsequently, Myb motif has been identified in a number of duplex telomere binding proteins in plants and yeasts [140,169,170]. It is worth mentioning that the TRF homolog in budding yeast, *Sc*TBF1, does not bind the duplex telomere DNA but binds subtelomeric TTAGGG repeats [179], using its C-terminal Myb domain [180]. On the other hand, budding yeast telomeres contain imperfect repeats [(TG_1-3_)_n_ in *S. cerevisiae* [76]] that are recognized by the RAP1 homologs [139]. Interestingly, *Sc*RAP1 also uses Myb-type DNA binding motifs to bind the duplex telomere DNA [140,141], although this was only revealed when the crystal structure of the *Sc*RAP1 central region was solved [140]. It turns out that *Sc*RAP1 has a central Myb domain and a Myb-Like domain that coordinate for duplex DNA binding [140]. Therefore, even though *Sc*RAP1 is not a homolog of mammalian TRFs, it still binds duplex telomere DNA using Myb motifs like TRFs. Both TRF and RAP1 homologs have been identified in *T. brucei* and are found to play important roles in VSG monoallelic expression and suppress VSG switching, which rely on their telomere DNA binding activities (see below).

Most known single-stranded telomere DNA binding proteins use OB folds to recognize the DNA [171], including hypotrichous ciliate *Oxytricha nova* TEBPα/TEBPβ [181,182,183,184,185], the human POT1/TPP1 heterodimer [115,186,187,188,189,190], and the CST complex [191,192,193,194,195,196,197,198,199,200], although the CST OB folds are different from the ones found in TEBP and POT1/TPP1 complexes [194], and CST has both sequence-specific and sequence-independent DNA binding activities [201,202,203]. Interestingly, although *T. brucei* telomere has a short single-stranded telomere G-overhang structure [84,85], no sequence-specific telomere G-overhang binding proteins have been identified, suggesting that *T. brucei* uses different protein(s) or mechanism(s) to protect the telomere termini.

### 2.3. T. brucei TRF and RAP1 Play Crucial Roles in Antigenic Variation

A number of telomere proteins have been identified in *T. brucei* (Figure 3) [19,55,204,205]. *Tb*TRF is the duplex TTAGGG repeat binding factor and a TRF homolog [204]. *Tb*RAP1 was identified as a *Tb*TRF-interacting factor and a RAP1 homolog [19]. *Tb*TIF2 interacts with *Tb*TRF and is a functional homolog of TIN2 [55]. In addition, TelAP1, PPL2, and PolIE were identified to be able to bind a DNA oligo with the telomere sequence [205]. Furthermore, *T. brucei* telomerase have been identified to be a major mechanism of telomere maintenance [206,207,208].

#### 2.3.1. The *Tb*TRF Myb Domain Has Sequence-Specific Duplex Telomere DNA and TERRA Binding Activities That Are Critical for Maintaining the Telomere Integrity and for Suppressing VSG Switching

Since *T. brucei* and vertebrates have the exact same telomere sequence, TTAGGG repeats [76], it is expected that telomere binding factors in *T. brucei* and vertebrates use similar DNA binding motifs. Indeed, *Tb*TRF was identified in silico, because the sequence of its C-terminal Myb domain is 33–38% homologous to those of mammalian TRF Myb domains [204]. TRF homologs have been identified in *T. cruzi* and *Leishmania*, two closely related kinetoplastid parasites [204,209]. While mammalian species have both TRF1 and TRF2, only one TRF homolog was identified in kinetoplastids despite an extensive sequence search [204,209].

*Tb*TRF associates with the telomere chromatin and co-localizes with the telomere throughout the cell cycle [204]. Knockdown of *Tb*TRF by RNAi leads to cell growth arrest and the loss of telomeric G-overhang, indicating that *Tb*TRF is essential for the terminal telomere structure and cell proliferation [204]. *Tb*TRF also self-dimerizes through a putative TRFH domain at its N-terminal region, a feature shared with other TRF homologs such as TRF1, TRF2, and *Sp*TAZ1 [178,204,210]. *Tb*TRF-depleted cells have an increased amount of DNA damage at the telomere [75] and an elevated VSG switching rate with many switching events involving the loss of the active ES [54], further indicating that defects in telomere integrity maintenance allow more VSG switching. Interestingly, depletion of *Tb*TRF leads to higher TERRA and TRL levels [75]. The R-loop structure has a tendency to introduce DNA damage [211,212,213]. Indeed, overexpression of RNase H1 [69] partially suppresses the elevated TRL level and the increased amount of DNA damage at the telomere in *Tb*TRF-depleted cells [75], indicating that suppressing the TERRA and TRL levels is an underlying mechanism of how *Tb*TRF helps maintain the telomere integrity.

*Tb*TRF uses its Myb domain to bind the duplex TTAGGG repeats in a sequence-specific manner [54,204]. Similar to the scenarios for human TRF1 and TRF2, the self-interaction of *Tb*TRF may enhance its potency and specificity for telomere binding. *Tb*TRF’s role in suppressing VSG switching relies on its telomere DNA binding activity, i.e., its Myb domain [54]. The structure of *Tb*TRF Myb domain, as determined by our team, is essentially identical to the Myb structures reported for other TRF homologs [54,214,215,216]. Specifically, the *Tb*TRF Myb domain adopts the canonical three-helix bundle architecture with the third helix predicted to insert into the major groove of telomere DNA for sequence-specific interactions [54]. NMR titration experiments and in vitro DNA binding studies led to the identification of several residues that are critical for the DNA binding activity of *Tb*TRF Myb domain, including R348 on the third helix that is conserved among TRF homologs and a few *Tb*TRF specific residues such as H346 and Q320 [54]. Mutational perturbations of these critical residues that completely abolish the DNA binding activity in vitro render *T. brucei* cells non-viable in vivo. The *Tb*TRF’s DNA binding activity is presumably essential for telomere integrity. Depletion of *Tb*TRF results in an increased amount of telomere DNA damage [75], and DNA damage at the active *VSG* vicinity results in lethality in >90% of cells [57]. Mutations that weaken but do not abolish the DNA binding activity, such as R298K, H346R, and R348K, yield cells that are viable with the VSG switching frequency elevated by ~1.6–3.3 fold [54]. These results confirm that the affinity of the *Tb*TRF Myb-DNA interaction is specifically important for suppressing the frequency of VSG switching [54].

Additionally, the *Tb*TRF Myb domain is found to bind to both regular and J-containing telomere DNA with similar affinity (*K*_d_ of 12 µM vs. 20 µM) [54]. J represents β-D-glucosyl(hydroxymethyl)uracil, a sugar-modified version of thymidine found in kinetoplastid flagellates [217]. In *T. brucei*, J is present only at the BF stage [218,219,220,221,222]. In kinetoplastids, only a small fraction (~1%) of thymidine bases in the genome are modified into J. Interestingly, J can be found in silent ESs [221] and is highly enriched in the telomere, replacing ~14% of T in (CCCTAA)_n_ and ~36% in (TTAGGG)_n_ [220,222]. It has been speculated that *Tb*TRF, as the only telomere DNA binding factor, may bind to the J base differently due to the bulky sugar added. However, the finding of comparable binding affinities suggests that the *Tb*TRF Myb domain does not differentiate between the T and J bases [54]. Structural modeling of the *Tb*TRF Myb domain in complex with J-containing telomere DNA also shows that neither the J base in the (CCCTAA)_n_ strand nor that in the (TTAGGG)_n_ strand are located close enough to the third helix of the Myb domain for possible direct interactions [54]. Thus, J base is unlikely to play a significant role in telomere integrity as it is not differentially recognized by the *Tb*TRF Myb domain. Instead, J base has been reported to regulate the transcription mediated by RNAP II as it is located at the ends of RNAP II PTUs [223,224,225].

The *Tb*TRF Myb domain is quite unique in that it can also bind TERRA [75]. Human TRF2 is also capable of binding TERRA and suppresses its level [226]. However, TRF2 uses its N-terminal basic GAR domain to bind TERRA [226,227] and TRF2 also promotes the TRL formation [228]. In contrast, *Tb*TRF suppresses the TRL level, and most interestingly, *Tb*TRF binds TERRA through its Myb domain [75]. The *Tb*TRF Myb domain also has a weak binding activity to CCCUAA repeats, but it clearly has a higher affinity to UUAGGG repeats [75]. Myb domain is well-known for its function to bind to dsDNA in a sequence-specific manner but has not been reported to possess binding activity for ssDNA or RNA [229]. Indeed, *Tb*TRF does not bind single-stranded telomere DNA [204]. Therefore, it is surprising that the *Tb*TRF Myb domain has a sequence-specific RNA binding activity, which is unique among all known Myb-containing telomere proteins. In addition, the R298E mutant abolishes the ds(TTAGGG)_n_ binding activity [54,75] but enhances the (UUAGGG)_n_ binding [75], suggesting that the DNA- and RNA-interacting interfaces overlap with each other (at least partially). Indeed, in vitro competition binding assays indicate that *Tb*TRF has a higher affinity to ds(TTAGGG)_n_ than (UUAGGG)_n_ [75]. Furthermore, the *Tb*TRF-DNA-RNA ternary complex has not been detected in in vitro analysis [75].

Overexpressing RNase H1 suppresses the increased amount of telomeric DNA damage and elevated TRL level in *Tb*TRF-depleted cells [75], indicating that suppression of the TRL level by *Tb*TRF is critical for telomere integrity and cell viability, as >90% cell die when a DSB is introduced in the active *VSG* vicinity [57]. It has been hypothesized that *Tb*TRF suppresses the TRL level through both its nucleic acid-binding activities. First, *Tb*TRF suppresses the TERRA level by telomeric silencing that presumably relies on its duplex telomere DNA binding activity [75]. Second, *Tb*TRF appears to promote trans-localization of TERRA [75], which may depend on both its TERRA and ds(TTAGGG)_n_ binding activities. It has been shown that *Tb*TRF-depleted cells have fewer nuclear TERRA foci, indicating that *Tb*TRF promotes trans-localization of TERRA [75]. Since *Tb*TRF binds both duplex telomere DNA and TERRA [75,204], and *Tb*TRF has a self-dimerization function [204], it is possible that *Tb*TRF recruits TERRA away from its transcription site, as a telomere-binding and a TERRA-binding TRF may interact with each other. The active ES is depleted of nucleosomes due to the high level of RNAP I transcription [27,28], and little *Tb*TRF is expected to bind the telomere that is transcribed by RNAP I. Therefore, TERRA is likely recruited away from its transcription site by *Tb*TRF, which helps reduce the local concentration of TERRA at the active telomere and limits the chance of TRL formation. The telomere DNA and TERRA binding activities of TRF homologs are summarized in Table 1.

#### 2.3.2. The Electrostatics-Based Sequence-Nonspecific dsDNA Binding Activity of *Tb*RAP1 Is Essential for VSG Silencing and Telomere Integrity

A yeast two-hybrid screen using *Tb*TRF as bait led to the identification of *Tb*RAP1, a homolog of yeast and mammalian RAP1s [19]. Based on sequence comparison with other RAP1 homologs, *Tb*RAP1 is predicted to contain an N-terminal BRCT domain, a Myb domain and a MybLike domain in the middle region, and a RAP1 C-Terminus (RCT) domain [19]. However, the exact boundaries for these domains are not well defined, due to the low sequence homology between *Tb*RAP1 and other RAP1 homologs. We have started to understand the key functions of these structural domains [22], but their precise functions still need further investigation.

*Tb*RAP1 is confirmed to be a *Tb*TRF-interacting factor by coimmunoprecipitation and shown to associate with the telomere by chromatin immunoprecipitation [19]. *Tb*RAP1 is also essential for *T. brucei* proliferation as depletion of *Tb*RAP1 by RNAi or conditional knockout of *Tb*RAP1 leads to cell growth arrest [19,20,21,22,63]. Most strikingly, knockdown of *Tb*RAP1 leads to derepression of all ES-linked silent *VSGs* [19,20,21,22]. Normally, only the active ES is colocalized with RNAP I at the extranucleolar ES body [230]. However, *Tb*RAP1 depletion leads to the formation of multiple extranucleolar RNAP I foci and simultaneous expression of multiple VSGs on cell surface [19]. Such derepression is specific to silent *VSGs* because the mRNA level of the active *VSG* was subtly decreased instead [19]. The *Tb*RAP1-mediated silencing effect spreads over the whole ES region and represses a reporter gene inserted immediately downstream of the ES promoter and 40–60 kb upstream of the telomere [19]. In addition, the *Tb*RAP1-mediated silencing effect is stronger for the telomere-adjacent *VSG* and weaker for the ES promoter-adjacent reporter gene, reflecting its position-dependent characteristic [19]. On the other hand, RNAP I-mediated transcription of rRNAs and RNAP II-transcribed genes like *Tb*TRF and histone H4 are not affected [19]. While *Tb*RAP1 knockdown led to profound derepression of all ES-linked silent *VSGs* [19,21,22], the molecular mechanism of this derepression is not completely understood. RAP1 homologs have been reported to repress the transcription of subtelomeric genes by strengthening the heterochromatic structure of the telomere [145,146,149,231,232,233]. *Tb*RAP1 also facilitates chromatin compaction at the telomere, at least in PF cells, and this appears to be a mechanism of *Tb*RAP1-mediated VSG silencing [20]. In *T. brucei*, histones [26,33,234], histone chaperones [235], a histone modifier [236], and chromatin remodelers [29,30,31] have been shown to play important roles in VSG silencing. However, whether any of these factors are recruited to the telomere by *Tb*RAP1 or other telomere proteins is unknown. Recent RNAseq analyses further showed that *Tb*RAP1′s role on gene expression is not only limited at the telomere [21,22]. *Tb*RAP1 depletion leads to an upregulation of more than 7000 genes, including essentially all *VSG* genes and many *ESAGs* (Figure 1) [21,22]. Interestingly, depletion of *Tb*RAP1 also causes a downregulation of more than 2500 genes, including many ribosomal protein genes [21,22]. Both yeast and mammalian RAP1s have been shown to regulate gene expression at non-telomeric loci [237,238,239,240,241,242,243]. It appears that *Tb*RAP1, likes its homologs, also has both activities to silence and activate gene expression at the telomere and non-telomere loci, respectively.

Association with the telomere chromatin is essential for the telomere functions of all RAP1 homologs. Interestingly, RAP1 homologs are recruited to the telomere through different means. Mammalian RAP1 does not bind the duplex TTAGGG repeat directly and is recruited to the telomere through its interaction with TRF2 [111,117]. Fission yeast RAP1 is also recruited to the telomere through its interaction with *Sp*TAZ1 [232], the fission yeast TRF homolog [111,141]. Although RAP1 homologs all have the central Myb domain, mammalian and fission yeast RAP1s do not possess any DNA binding activity, and their Myb domains show sequence and structural features that are incompatible with direct DNA binding [244].

Recently, our team reported that a stretch of positively charged residues (_737_RKRRR_741_) in *Tb*RAP1 is responsible for its unique DNA binding activities [21]. This R/K-patch is located within the MybLike domain and overlaps with *Tb*RAP1′s nuclear localization signal (aa 727 to 741) [21]. In vitro biochemical studies confirm that this R/K-patch allows *Tb*RAP1 to bind to both single- and double-stranded DNA in an electrostatics-based, sequence-nonspecific, and substrate size-dependent manner [21]. This R/K-patch is also required for *Tb*RAP1′s localization to the telomere while *Tb*TRF and the Myb domain of *Tb*RAP1 are dispensable [21]. Interestingly, the dsDNA binding activity of *Tb*RAP1 mediated by the R/K-patch appears to be sensitive to the phosphorylation status of S742 and S744 [21], two residues adjacent to this positively charged segment with their phosphorylated state detected in both BF and PF cells [245,246]. Phosphomimicking mutations S742D/S744D disrupt the telomere localization of *Tb*RAP1, causing telomere damage and derepressed ES-linked silent *VSGs* [21]. Another *Tb*RAP1 mutant, with the positively charged R/K-patch mutated to _737_AAAAA_741_, causes the same phenotypes [21]. Thus, the electrostatics feature of the R/K-patch is of critical importance to the unique DNA binding activity of *Tb*RAP1, which is essential for telomere integrity and *VSG* silencing [21]. The telomere DNA binding activities of RAP1 homologs are summarized in Table 2.

## 3. Conclusions

Antigenic variation is a key pathogenesis mechanism of *T. brucei,* enabling the parasite to establish long-term infections and rendering vaccination ineffective [247]. Work from our and others’ laboratories have shown that the telomere structure and telomere proteins are both critical for monoallelic VSG expression and affect VSG switching frequencies [19,20,21,22,54,55,56,57,61,63,68,75].

The overall telomere architecture is conserved between *T. brucei* and higher eukaryotes. Homologs of several Shelterin components have also been identified in *T. brucei*, including *Tb*TRF and *Tb*RAP1 [19,55,204]. Although *Tb*TRF and *Tb*RAP1 have conserved telomere functions as their respective homologs, the underlying mechanisms are not the same [19,21,54,75,204]. As described above, the *Tb*TRF Myb domain binds to TERRA in addition to ds(TTAGGG)_n_ [54,75,204], making this Myb domain unique among known telomere protein Myb domains [216], bearing both sequence-specific dsDNA and RNA binding activities. Similarly, *Tb*RAP1′s unconventional electrostatics-based, sequence-nonspecific DNA binding activities are also unique among all RAP1 homologs [21]. Importantly, the DNA binding activities of *Tb*TRF and *Tb*RAP1 are required to maintain the telomere integrity (hence cell viability) and suppress VSG switching and are essential for monoallelic VSG switching [21,54].

Given the importance of their unique DNA binding activities, *Tb*TRF and *Tb*RAP1 can serve as potential targets to develop effective therapeutics against *T. brucei*. For example, molecular agents that interfere or abolish the DNA binding activities of *Tb*TRF and *Tb*RAP1 may reduce *T. brucei* viability by compromising telomere integrity and weakening the defense mechanism of antigenic variation through loss of VSG monoallelic expression. Such a drug discovery effort may bring enormous health and economic benefits to those disadvantaged regions exposed disproportionately to the risk of this parasite. In addition, as *T. cruzi* and *Leishmania* are closely related to *T. brucei*, and TRF and RAP1 homologs are readily identifiable in these parasites [19,204,209], knowledge gleaned from studies on *T. brucei* TRF and RAP1 will also help fight *T. cruzi* and *Leishmania* infections.

## Figures and Tables

**Figure 1 pathogens-10-00967-f001:**
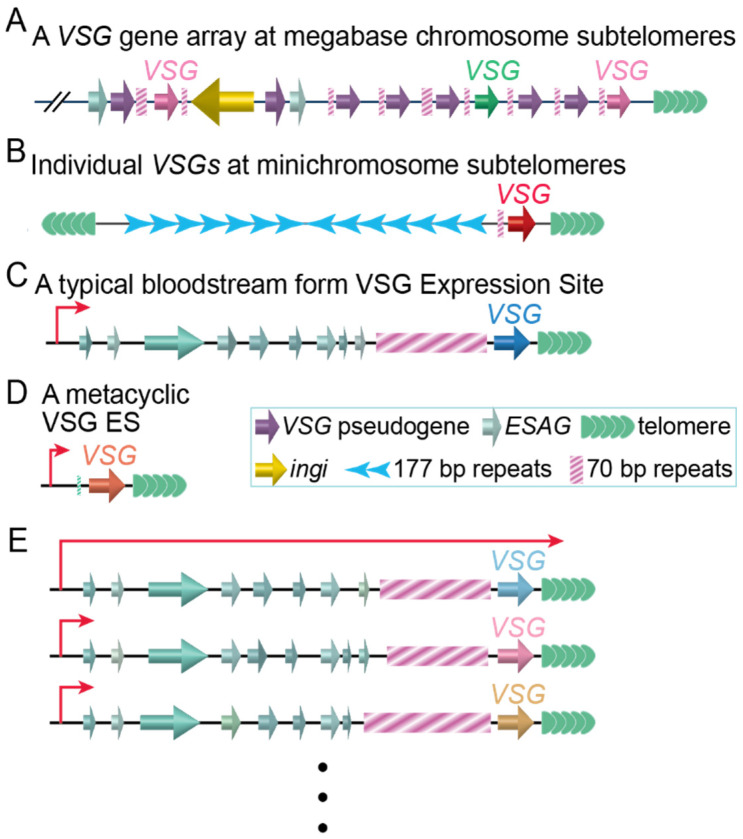
Representative *VSG* loci in *T. brucei*. (**A**) A *VSG* gene array. (**B**) A minichromosome with a subtelomeric *VSG* gene. (**C**) A typical BF VSG ES. *ESAG*: *ES-Associated Gene.* (**D**) A typical metacyclic VSG ES. (**E**) Only one VSG ES is fully active at any time. Dots at the bottom represent multiple similar BF ESs of which details are not shown.

**Figure 2 pathogens-10-00967-f002:**
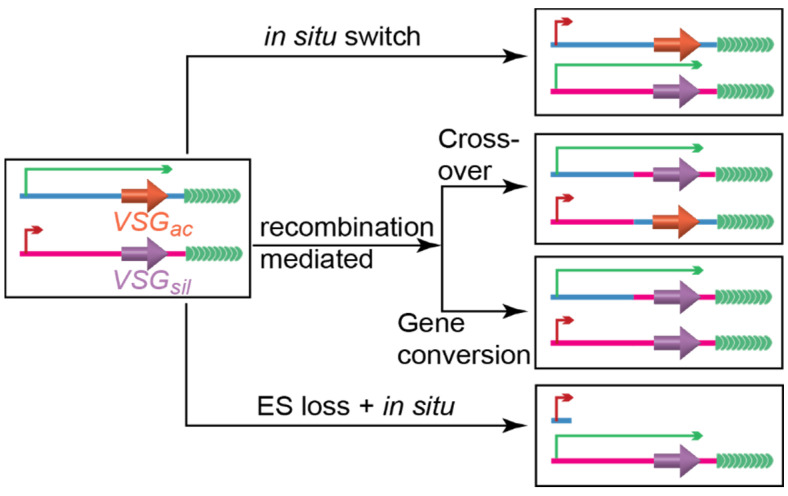
Major VSG switching pathways. *VSG_ac_*, the active *VSG*; *VSG_sil_*, a silent *VSG*. Long green arrow represents the active ES promoter, and short red arrow represents a silent ES promoter.

**Figure 3 pathogens-10-00967-f003:**
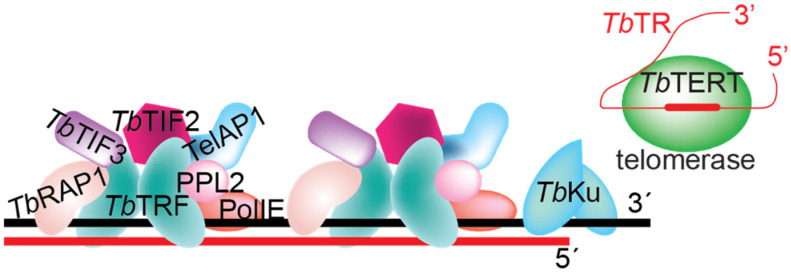
The *T. brucei* telomere protein complex. Core telomere protein components are shown.

**Table 1 pathogens-10-00967-t001:** Summary of nucleic acid binding activities of TRF homologs.

	TRF Homologs						
Species	Protein	DNA Binding		TERRA Binding		Homodimerization	
		Duplex Telomere DNA Binding	Binding Domain	TERRA Binding	Binding Domain	Activity	Domain
Human	hTRF1	binds ds(TTAGGG)_n_	C-terminal Myb	Not reported	N/A	Yes	TRFH
	hTRF2	binds ds(TTAGGG)_n_	C-terminal Myb	binds (UUAGGG)_n_	N-terminal GAR domain	Yes	TRFH
Budding yeast	*Sc*TBF1	binds ds(TTAGGG)_n_	C-terminal Myb	Not reported	N/A		
Fission yeast	*Sp*TAZ1	binds ds[G_2–8_TTAC(A)]_n_	C-terminal Myb	Not reported	N/A	Yes	
	*Sp*TBF1	binds ds[G_2–8_TTAC(A)]_n_	C-terminal Myb	Not reported	N/A		
*T. brucei*	*Tb*TRF	binds ds(TTAGGG)_n_	C-terminal Myb	binds (UUAGGG)_n_	C-terminal Myb	Yes	TRFH

**Table 2 pathogens-10-00967-t002:** Summary of DNA binding activities of RAP1 homologs.

		RAP1 Homologs	
Species	Protein	DNA Binding	DNA Binding Domain
Human	hRAP1	No	N/A
Budding yeast	*Sc*RAP1	binds ds(TG_1–3_)_n_	Myb and Myb-Like
Fission yeast	*Sp*RAP1	No	N/A
*T. brucei*	*Tb*RAP1	binds ds- & ssDNA	_737_RKRRR_741_

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
