# Peer review of "Regulation of Antigenic Variation by Trypanosoma brucei Telomere Proteins Depends on Their Unique DNA Binding Activities"

_pathogens, 2021, doi:10.3390/pathogens10080967_

Round 1

Reviewer 1 Report

This is a well-written and comprehensive review about an important topic of trypanosome biology, namely the functional roles of telomeric proteins in antigenic variation, the parasite's main immune evasion strategy.

Two major comments.

(1) The two figures about VSG expression sites (ESs) in this review have been published in various versions numerous times starting in the 1980s. This, however, is a review about telomeric proteins and not ESs. Hence, the manuscript would strongly benefit from a figure depicting the telomeric proteins discussed and their telomeric interactions [with ESs].

(2) A key concept of telomeric proteins and their role in telomeric silencing, i.e. the repression of VSG ESs, is that they recruit histone modifiers/remodellers and spread heterochromatin inward. To my knowledge at least DOT1B was identified as such an enzyme involved in telomeric silencing in trypanosomes - it is not mentioned. However and according to the title, the authors need to discuss the knowledge around which 'silencing' proteins may be recruited by the telomeric proteins.

This should be considered though.

Author Response

Response to reviewer 1

We thank the reviewer for the comments and suggestions. Specifically,

  • We agree that similar diagrams of VSG ESs and VSG switching have been shown in many review articles. However, to help non-tryp readers to better appreciate the mechanisms of VSG regulation, we think it’s beneficial to keep both figure 1 and figure 2. We also agree that a diagram of the brucei telomere complex is helpful, and we have added figure 3 to show the T. brucei telomere protein complex as we know it currently.
  • We thank the reviewer for the suggestion. We have added that histones, histone chaperones, a histone modifier, and chromatin remodelers are important for VSG silencing, but it is unknown whether any of these factors are recruited to the telomere by telomere proteins (page 15, last paragraph).

Reviewer 2 Report

Trypanosoma brucei uses antigenic variation of it major surface protein, the VSG, to persist and thrive in its mammalian host. Telomere proteins play a role in regulating antigenic variation which is tightly controlled. This is a nice review by Li and Zhao that focuses on the functions of two of these known telomere binding proteins, TbTRF and TbRAP1, and explores their use as potential targets in therapeutic approaches.

I have a few minor suggestions:

  • In the last paragraph of section 1 on page 2 the life cycle of brucei is described. It might be worth pointing out here that this is not a comprehensive description as not all life cycle stages are mentioned.
  • In Fig. 1 ‘VSG pseudogene’ and ‘ESAG‘ are difficult to distinguish with the chosen color scheme. A change in colors would be helpful.
  • The caption of Fig. 1 reads the following for C: A tropical BF VSG ES. Correct to read typical.
  • In conclusions the colloquial term ‘labs’ is used in line 3. Exchange for laboratories.
  • There are a few minor spelling mistakes such as incorrect capitalization of words, e.g. Polymerase, that should be corrected.

Author Response

Response to Reviewer 2

We thank the reviewer for the comments and suggestions. Specifically,

  • We have revised the text to state that only key life cycle stages are described (page 3, 2nd paragraph)
  • We have changed the color scheme for VSG pseudogene and ESAG in figure 1.
  • We have corrected the figure 1C legend.
  • We have changed “labs” to “laboratories”.
  • We have corrected “polymerase” throughout the text.